# CONTRASTIVE META LEARNING FOR DYNAMICAL SYSTEMS

## ABSTRACT

Recent advancements in deep learning have significantly impacted the study of dynamical systems. Traditional approaches predominantly rely on supervised learning paradigms, limiting their scope to large scale problems and adaptability to new systems. This paper introduces a novel meta learning framework tailored for dynamical system forecasting, hinging on the concept of mapping the observed trajectories to a system-specific embedding space which encapsulates the inter-system characteristics and enriches the feature set for downstream prediction tasks. Central to our framework is the use of contrastive learning for trajectory data coupled with a series of neural network architecture designs to extract the features as augmented embedding for modeling system behavior. We present the application of zero-shot meta-learning to dynamical systems, demonstrating a substantial enhancement in performance metrics compared to existing baseline models. A notable byproduct of our methodology is the improved interpretability of the embeddings, which now carries explicit physical significance. Our results not only set a new benchmark in the field but also pave the way for enhanced interpretability and deeper understanding of complex dynamical systems, potentially opening new directions for how we approach system analysis and prediction.

## 1 INTRODUCTION

In the field of dynamical systems, the application of deep learning represents a significant development, introducing new perspectives and methods for analyzing complex temporal behaviors. Historically, the analysis and prediction of these systems have largely depended on supervised learning techniques (Kumpati et al., 1990; Ljung, 1998; Hefny et al., 2015; Brunton et al., 2016). Although these techniques have proven useful in specific situations, they have certain drawbacks, particularly in terms of applying acquired knowledge to novel or changing systems. Supervised learning models typically necessitate accurately labeled datasets tailored to each individual system, resulting in a lack of applicability to scenarios involving extensive volumes of unlabeled data and flexibility to transfer knowledge to new systems. This specific adaptation to particular datasets poses generalization issues (Kirchmeyer et al., 2022), as the models face difficulties in adjusting to the new systems with varying dynamics or conditions. Moreover, the reliance on substantial volumes of labeled data renders these models impractical in situations where data is costly or less feasible to acquire, or in instances where systems evolve beyond the distribution of the initial training data.

Meta-learning (Huisman et al., 2021; Hospedales et al., 2021) and multitask learning (Zhang & Yang, 2018; Caruana, 1997) approaches have gained traction for their ability to handle multiple tasks and adapt to new scenarios. These approaches leverage "comparison" as a fundamental concept (Tian et al., 2020b), utilizing it to derive embeddings that uniquely identify objects across different categories. This methodology is exemplified by innovations such as the siamese network (Koch et al., 2015) and triplet loss (Schroff et al., 2015), which paved the way for the emergence of "contrastive learning". This branch of machine learning, together with the pre-train and fine-tune strategies, facilitates few-shot or zero-shot learning, enabling models to apply knowledge from extensive datasets to previously unseen data. Despite their successes, applying these methodologies directly to dynamical systems poses substantial hurdles (Nagabandi et al., 2018; Wang et al., 2022), due to the unique patterns of such systems. Unlike traditional meta-learning where the new tasks or unseen data are typically defined in categorical terms, the concept of embedding in dynamical sys-

tems lacks the notion of distinct classes. This absence of clear-cut categories presents a challenge in adapting existing contrastive learning methods to the field of dynamical systems.

Motivated by this challenge, our paper introduces a specialized meta-learning framework tailored for dynamical systems. This framework encompasses a dual-phase process, firstly focusing on the discovery of unique system embeddings by contrastive learning and subsequently employing these embeddings for forecasting tasks. Our contributions are summarized as follows:

1. We introduce a novel perspective contrastive learning applied to dynamical system identification problems, by comparing truncated trajectories sampled both within a single system (intra-system) and across different systems (inter-system) to learn an effective representation of the system dynamics. To our knowledge, this study is the first exploration of zero-shot meta-learning techniques for dynamical systems that does not require adaptation to new systems or explicit labeling of system-specific coefficients.

2. We systematically developed a learning framework for meta dynamical system learning, incorporating distinctively designed modules. These include the "Local Linear Least Square" feature extractor for vector-based systems and "Spatial Adaptive LinEar Modulation (SALEM)" for grid-based system. Furthermore, we also proposed a dimensional square ratio contrastive loss function, uniquely tailored for trajectory contrastive learning in dynamical systems.

3. Through the synthetic experiments, we offer qualitative evidence of the efficacy of contrastive learning in the embedding space. We also quantitatively demonstrate that the forecasting errors with learned embedding are significantly lower compared to those of baseline neural networks in dynamics prediction.

## 2 BACKGROUND AND RELATED WORK

### 2.1 DEEP LEARNING BASED DYNAMICAL SYSTEM LEARNING

In the context of identifying and understanding dynamical systems, deep learning has emerged as a popular and powerful tool at handling the temporal dependencies and nonlinear dynamics characteristic of these systems. Earlier models such as recurrent neural networks (RNNs) (Bailer-Jones et al., 1998) excel in capturing these intricate temporal patterns, enabling predictions and analyses of discrete time system behaviors. Following the setup of continuous-time dynamical systems, the concept of Neural Ordinary Differential Equations (Neural ODEs) has been introduced by Chen et al. (2018). For predicting complex discrete time system such as Partial differential equations (PDEs), ResNet (He et al., 2016) is commonly used as backbone model for forecasting the future dynamics (Long et al., 2018; 2019; Xu et al., 2019).

### 2.2 META DYNAMICAL SYSTEM LEARNING

Meta dynamical system learning, being a merging branch of dynamical system learning, focuses on developing models that can adapt and generalize across various physical systems and environments. Yin et al. (2021) presents LEADS, a new framework for modeling multiple dynamical systems, by capturing common dynamics within a shared model while also accounting for environment-specific model. Following the work, Kirchmeyer et al. (2022) introduces a hypernetwork learned jointly with a context vector from observed data, aims for fast adaptation and enhanced generalization across environments with minimal data samples. Wang et al. (2022) proposed DyAd, a meta-learning framework comprising an encoder that deduces time-invariant hidden features of the task with limited supervision, and a forecaster that generalizes the dynamics of the entire domain. More recently, Blanke & Lelarge (2023) proposes CAMEL, parameterizing an affine structure to accommodate the new task by few shot learning. Nevertheless, these earlier meta-learning models for dynamical systems depended on supervised training on labeled system coefficients or few shot adaptation to new systems, limiting their usability in complex situations where environments rapidly change, or prior knowledge about the new systems is unavailable. Our approach, on the other hand, employs contrastive learning to automatically identify system coefficients without relying on labeled data. By leveraging unsupervised learning techniques and zero-shot forecasting, our method can operate effectively even in the absence of prior knowledge about new systems.

## 2.3 Contrastive Learning

Contrastive Learning, diverging from traditional supervised learning methods that directly map data, focuses on implicitly deriving data representations by comparing examples. Its inception dates back to the early 1990s, as evidenced by foundational works such as Bromley et al. (1993). The method has been extensively applied across various domains, significantly impacting metric learning (Chopra et al., 2005; Sohn, 2016), a field closely related to our work.

In the community of unsupervised and semi-supervised learning, contrastive learning has gained prominence, especially in self-supervised learning (SSL) tasks. Its effectiveness is well-established in areas such as computer vision and natural language processing, as demonstrated in research by Chen et al. (2020a); He et al. (2020); Tian et al. (2020a) for computer vision, and Wu et al. (2020); Gao et al. for natural language processing. There has been notable advancements in the design of contrastive loss, with significant contributions from Oord et al. (2018a); Chen et al. (2020a;b).

In recent years, its application to time series data has gained attention due to the unique challenges posed by the temporal nature of the data (Pöppelbaum et al., 2022; Yue et al., 2022), enabling tasks such as anomaly detection, clustering, and classification. Despite the promising results in the time series domain, which is closely related to dynamical systems, the application of contrastive learning to the latter remains largely unexplored. The major challenges include the requirement for high interpretability in dynamical systems, as they are often governed by physical laws and principles, and the need for learned representations to align with these underlying physical mechanisms. Additionally, the diverse forms of state representation in dynamical systems, such as grid-based systems (to be introduced later), pose difficulties in integrating learned embedding into downstream tasks, requiring careful consideration of how the representations can be effectively utilized.

## 3 Problem Formulation

### 3.1 Representation Learning of Multiple Dynamical Systems

In this paper, we consider two types of common settings of autonomous dynamical systems: continuous time and discrete time systems. In the former, we consider the unknown system equation in the following form: $\dot{x} = dx(t)/dt = f(x)$, where $x \in \mathcal{X}$ is the system state and $\dot{x}$ is its time derivative to time index $t$, and the future state depends solely on the current state if function $f(\cdot)$ is fixed. In the discrete time step setting, we consider the autonomous system equation in the form of $x_{t+1} = f(x_t)$[1]. $f \in \mathcal{F} : \mathcal{X} \to \mathcal{T}\mathcal{X}$ or $\mathcal{X}$ denotes the system function that maps system state space $\mathcal{X} \in \mathbb{R}^n$ to its temporal derivative space $\mathcal{T}\mathcal{X} \in \mathbb{R}^n$ or future state space $\mathcal{X}$.

$\mathcal{F}$ is the functional space that $f(\cdot)$ belongs to, and we consider that all functions $f(\cdot)$ share certain characteristics but may vary in some aspects. The difference can be represented by a set of parameters $\phi$ (e.g., physics constants, material properties), that is $f_\phi(\cdot) \in \mathcal{F}$, where $\phi \in \Phi$ and $\Phi$ is the coefficient space. The main objective is to develop methodologies to establish an encoder from a trajectory observation to an informative embedding corresponding to $\phi$ without prior knowledge of it. By doing so, we can accurately parameterize $f_\phi(\cdot)$ and employ it to forecast the evolution of future trajectories.

### 3.2 Generalizing to new dynamical systems

Under this framework, we construct a structured representation for this composite dynamical system, composed of a set of dynamical sub-systems $\{f_{\phi_i}(\cdot)\}$, each characterized by a specific coefficient or physics properties $\phi_i$. Take the continuous time system for example, given an initial condition $x_0$, the trajectory is in the form of:

---

[1] While dynamical systems typically depend solely on the current state, practical scenarios may lack complete state information. For instance, PDE solutions might omit boundary conditions. We therefore relax the single-state assumption, framing the task as a mapping that considers multiple time steps: $f : (x_{t-t_i+1}, ..., x_t) \to (x_{t+1}, ..., x_{t+t_o})$, where $t_i$ and $t_o$ are input and output time steps, respectively. This approach leverages past observations to predict multiple future states, we follow the same setup from Wang et al. (2022) in the experiments of grid-based systems.

$$x(t) = x_0 + \int_0^t f_{\phi_i}(x(\tau))d\tau$$

For each sub-system characterized by the function $f_{\phi_i}(\cdot)$, we may have one or more observed trajectories of the system states. These trajectories are then sampled, resulting in a set of data points denoted as $\{x_t^{i,j}\}_{t=0}^T$. The indices $i$, $j$, and $t$ represent the system coefficients, initial conditions, and time steps, respectively.

In the context of meta dynamical system learning, our goal is to develop a model that can generalize from a set of training systems $\phi_{\text{train}}$ to new, unseen test systems with properties $\phi_{\text{test}}$. This presents a challenging task of learning a meta-model that captures the underlying principles governing these dynamical systems across different coefficient spaces. Moreover, given a new system, we would like to leverage a short observed trajectory to infer its system embedding, with which we can augment to accurately forecast its future evolution. This approach enables us to quickly adapt our model to previously unseen systems, making predictions based on limited observations while leveraging the knowledge gained from the training systems.

## 4    PROPOSED METHOD

This section presents our two-step meta-learning framework tailored for dynamical systems, as shown in Figure 1. The first step involves employing contrastive learning to derive embeddings for each trajectory, capturing the coefficients or the physics properties of the dynamical system. In the second step, we apply the embedding model to the initial segment of the trajectory to deduce the embedding. These embeddings, representing inferred coefficients, serves as inputs for dynamical system model to predict the future trajectories.

In the experimental section, we evaluate the framework on two types of dynamical systems: continuous-time vector-based systems with fully observable states and interpretable physics coefficients, and grid-based, multi-channel systems (e.g. PDE solution represented on spatial grids for each state) in discrete time steps. The remaining sections are organized as follows: Section 4.1 introduces the contrastive loss design, Section 4.2 and Section 4.3 discuss the design of the "encoder" and dynamical system learning for continuous-time vector-based systems, and Section 4.4 describes the "decoder" design to incorporate trajectory knowledge for grid-based discrete time dynamical systems.

### 4.1    CONTRASTIVE LEARNING DESIGN FOR TRAJECTORY EMBEDDINGS

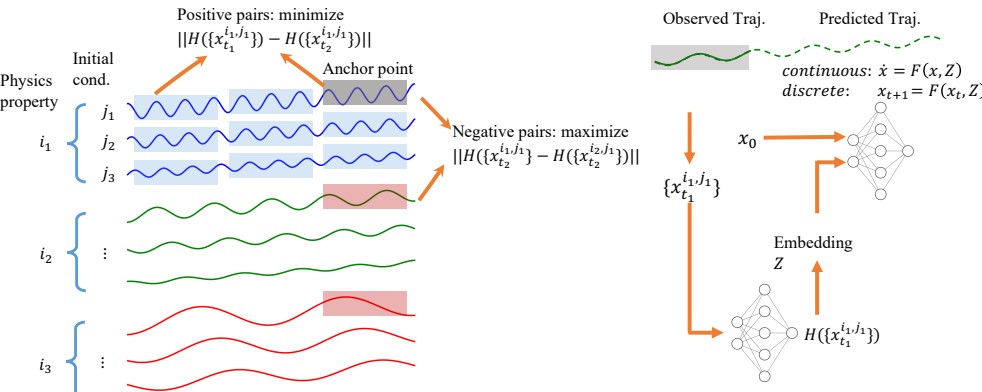

Figure 1: Two-step learning pipeline for multiple dynamical systems: 1: employing a contrastive learning framework to extract system embeddings from trajectory observations, 2: using inferred embeddings for dynamic model forecasting of future trajectories.

At its core, contrastive learning involves training a model to distinguish between similar ("positive") and dissimilar ("negative") pairs of data samples. In the context of trajectory observation of multiple

dynamical systems, positive pairs are truncations of trajectories that belong to the same sub-system, suggesting similar underlying state evolution patterns, while negative pairs are truncations that are drawn from different sub-systems. Inspired by the Square Ratio Loss (SRL) from Zhang et al. (2023), we propose a new Element-wise Square Ratio Loss (ESRL):

$$\mathcal{L}_{\text{ESR}} = \frac{1}{MD} \sum_{i_1,j_1,t_1} \left[ \sum_{d=1}^{D} \frac{\sum_{j_2,t_2} (H_{\theta_{tr}}^d(\{x_{t_1}^{i_1,j_1}\}) - H_{\theta_{tr}}^d(\{x_{t_2}^{i_1,j_2}\}))^2}{\sum_{i_3,j_3,t_3} (H_{\theta_{tr}}^d(\{x_{t_1}^{i_1,j_1}\}) - H_{\theta_{tr}}^d(\{x_{t_3}^{i_3,j_3}\}))^2} \right]. \tag{1}$$

where $H_{\theta_{tr}(\cdot)}$ is the neural network parameterized by $\theta_{tr}$, to map a trajectory to $d$-dimensional embedding space. $H_{\theta_{tr}}^d(\cdot)$ refers to the $d^{\text{th}}$ dimension or element of this mapping. Each trajectory, denoted as $\{x_{t_1}^{i_1,j_1}\}$, is identified with system coefficient index $i_1$, trajectory truncation index $j_1$ and starting time index $t_1$. For brevity, we omit the specifics of the time index range. $M$ is the number of anchor points for normalization purpose.

On the numerator of Equation (1), we compare the element-wise square distance between the trajectories drawn from same set of coefficients. Conversely, the denominator accounts for all possible pairs of trajectories within the mini-batch. In practice, for each set of coefficients $\phi_i$, multiple trajectories are generated, each originating from a unique set of initial conditions. We then create input sequences by randomly truncating these trajectories into shorter segments. These truncated sequences serve as individual inputs to the encoder in Equation (1).

The Square Ratio Loss introduced by Zhang et al. (2023), along with contrastive loss like the Triplet (Schroff et al., 2015) and Info-NCE loss [2] Oord et al. (2018b); Chen et al. (2020a), traditionally employ vector-based Euclidean distance in their ratio loss functions. During contrastive learning, minimizing this loss function will decrease the distance between positive pairs and increase the discrepancy between negative pairs. However, this setup could result in trivial solutions in multi-dimensional embeddings. Specifically, embeddings might vary across certain dimensions but remain constant in others (e.g. learned embeddings for 3 different systems are represented as $\{[1,0],[2,0],[3,0]\}$, the 2nd dimension becomes constant). These invariant dimensions are ineffective for training as they do not contribute to the loss function. To address this issue, we propose the element-wise ratio loss in Equation (1), which penalizes the loss if certain dimensions of the embedding become constant.

Further, there could exist other dimensional collapse such that the two embedding dimensions became correlated (e.g. learned embeddings for 3 different systems are represented as $\{[1,2],[2,4],[3,6]\}$, the two dimensions are linearly correlated). To avoid dimensional collapse of the embedding dimensions, we introduce a correlation regularizer inspired by the covariance regularizer from Bardes et al. (2021).

Given $z_{n,d}$ as the $d_{th}$ dimension of the embedding from the $n_{th}$ trajectory truncation from the mini-batch, let $\tilde{z}_{n,d} = (z_{n,d} - mean(\{z_{n',d}\}_{n'=1}^N))/\sqrt{var(\{z_{n',d}\}_{n'=1}^N)}$ be the normalized feature, where $mean(\cdot)$ and $var(\cdot)$ indicates the mean and variance over the dimension of truncations. Let $\tilde{Z} = \{\tilde{z}_{n,d}\}_{n,d=1,1}^{N,D}$ be the normalized matrix. The correlation matrix $C$ is in the following form and the regularizer penalizes all the non-diagonal elements of the correlation matrix if they get close to $\pm 1$:

$$C(\tilde{Z}) = \tilde{Z}^\top \tilde{Z} \tag{2}$$

$$\mathcal{L}_{\text{cov}} = \frac{1}{D-1} \sum_{i \neq j} [C(\tilde{Z})_{i,j}^2] \tag{3}$$

In practice, we use the following total loss as a summation of the prior two losses. The coefficient $\lambda$ is set to 0.5 across all the standard experiments:

$$\mathcal{L}_{\text{total}} = \mathcal{L}_{\text{ESR}} + \lambda \mathcal{L}_{\text{cov}} \tag{4}$$

---

[2] The cosine similarity, which is equivalent to the inner product of normalized embeddings, is inversely related to the Euclidean distance between these normalized vectors.

One might wonder why the more popular choice of contrastive loss functions (Oord et al., 2018a; Chen et al., 2020a;b; Schroff et al., 2015) are not employed in this case. Firstly, they are subjected to the dimensional collapse issue above. Secondly, these probability-based loss functions may be better effective for classification tasks, while may not be as suited for dynamical system applications, where the distribution of embeddings is more continuous in nature. A more detailed discussion of the above topics with corresponding comparison study are provided in Appendix C.1.

## 4.2 LOCAL LINEAR LEAST SQUARE FEATURE EXTRACTOR FOR VECTOR-BASED SYSTEMS

Recurrent neural networks (RNNs) is a natural choice for handling the sequential inputs of the dynamical systems. However, we found that directly applying RNNs to vector-based trajectory data is not sufficient for extracting the underlying physics of the dynamical systems (example learned embedding space shown in Appendix C.1). To address this, we introduce a transformation layer to pre-process the trajectory inputs for better extraction. Given a trajectory $\{x_t\}_{t=1}^T \in \mathbb{R}^{T \times D}$, we first split it into multiple segments with length $r$:$\{x_t\}_{t=1}^r, \{x_t\}_{t=r}^{2r} \dots \{x_t\}_{t=T-r+1}^T$. We then model each segment under the assumption that the dynamical system is locally linear (i.e., follows $\dot{x} = Ax$), and calculate the optimal $A \in \mathbb{R}^{D \times D}$ that best fits the data. It is important to note that the concept of "segment" in this context should not be confused with the trajectory "truncations" used in contrastive learning.

$$\hat{A}_k = \arg\min_A \sum_{t=1+(k-1)r}^{kr} ||Ax_t - \hat{\dot{x}}_t||^2 \tag{5}$$

where $\hat{\dot{x}}_t$ is the time derivative of $x_t$ approximated by finite difference methods (i.e. $\hat{\dot{x}}_t = (x_t - x_{t-1})/\Delta t$). After extracting $\{\hat{A}_k\}_{k=1}^{T//r}$, each $\hat{A}_k \in \mathbb{R}^{D \times D}$ is reshaped into a vector in $\mathbb{R}^{D^2 \times 1}$. This vector is then used as step input for the embedding network.

This preprocessing step transforms the original trajectory into a sequence of locally linear approximations, which can be more easily processed by the RNN to extract meaningful embeddings about the underlying dynamical system. We can think of this extractor as a tool that captures the non-linear system as a series of linear "snapshot" of the system's behavior at different points. An RNN then processes this sequence of coefficients, converting them into a compact embedding that represents the system's overall behavior. After the preprocessing step, the resulting sequence of locally linear approximations still needs to be divided into truncations for the purpose of contrastive training at a later stage.

## 4.3 DYNAMICAL SYSTEM LEARNING FOR CONTINUOUS TIME SYSTEMS

Once the trajectory embedding/encoder network is fully trained, it is integrated with the forecaster neural network. For the vector-based systems, we simply use a multi-layer perceptron (MLP) which takes in the concatenation of the system state inputs and the trajectory embedding. For each trajectory in the dataset, the initial segment is utilized to calculate the trajectory embedding, while the subsequent segment is employed for training the dynamics. With the forecasting neural network denoted as $F_{\theta_d}(\cdot)$ and trajectories represented by $\{x_t^{i,j}\}_{t=1}^T$, the loss function is defined as:

$$\mathcal{L}_{\text{continuous}} = \frac{1}{M(T-s+1)} \sum_{t=t_{s+1}}^T \sum_{i,j} ||\dot{x}_t^{i,j} - F_{\theta_d}(x_t^{i,j}, H_{\theta_{tr}}(\{x_{t'}^{i,j}\}_{t'=t-s+1}^t))||^2, \tag{6}$$

where $\{x_{t'}^{i,j}\}_{t'=t-s+1}^t$ is the observed trajectory ahead of timestamp $t$ with length $s$ to infer the embedding. For simplicity, we assume the time derivative of the system state $\dot{x}_t^{i,j}$ is known as data. If it is unknown, we can also use finite difference to estimate it or integrate the ODE to compare with discrete time observations.

## 4.4 SPATIAL ADAPTIVE LINEAR MODULATION (SALEM) FOR DISCRETE-TIME SYSTEMS LEARNING

For forecasting (2D) grid-based systems, ResNets are commonly used as baseline backbone models (Long et al., 2018; 2019; Xu et al., 2019). Typically, the model takes an input sequence shape

of $[T, C, H, W]$ and outputs a future step represented as $[1, C, H, W]$, where $T$ is the number of past observation steps, $C$ is the channel number and $H, W$ are the grid image height and width respectively. When forecasting multiple steps into the future, an auto-regressive iterative approach is necessary. For training discrete-time dynamical systems, the loss function in Equation (7) employs the average squared difference between predicted and actual next state, rather than using time derivative components in Equation (6). For evaluation, multiple time stamp are forecasted with auto-regressive manner.

$$\mathcal{L}_{\text{discrete}} = \frac{1}{M(T-s+1)} \sum_{t=t_{s+1}}^{T} \sum_{i,j} ||x_{t+1}^{i,j} - F_{\theta_d}(\{x_{t'}^{i,j}\}_{t'=t-s+1}^{t}, H_{\theta_{tr}}(\{x_{t'}^{i,j}\}_{t'=t-s+1}^{t}))||^2, \quad (7)$$

To incorporate the learned vector embedding into the grid-based prediction, we draw inspiration from two techniques: Feature-wise Linear Modulation (FiLM) by Perez et al. (2018) and Spatially-Adaptive Normalization (SPADE) from Park et al. (2019). In FiLM, the channel values are modulated by a single vector, while in SPADE, the normalization parameters are conditioned on spatial semantic information.

We propose a new modularization using both vector embedding and spacial information, named as Spatially Adaptive LinEar Modulation (SALEM), as shown in Figure 2. In the SALEM block, we set up an arbitrary spacial coordinate system for the domain ($x$-$y$), and each coordinate will be concatenated with the embedding $Z$ to be mapped to a vector pair of $\{(\gamma_{c,x,y}, \beta_{c,x,y})\}_{c=1,2...C}$, where $(\gamma_{c,x,y}, \beta_{c,x,y})$ modulates the feature value at $(x, y)$ in the $c_{th}$ channel by performing the following affine transformation:

$$\gamma_{c,x,y} = m_1(Z, x, y), \beta_{c,x,y} = m_2(Z, x, y) \quad (8)$$
$$SALEM(F_{c,x,y}|\gamma_{c,x,y}, \beta_{c,x,y}) = \gamma_{c,x,y} * F_{c,x,y} + \beta_{c,x,y} \quad (9)$$

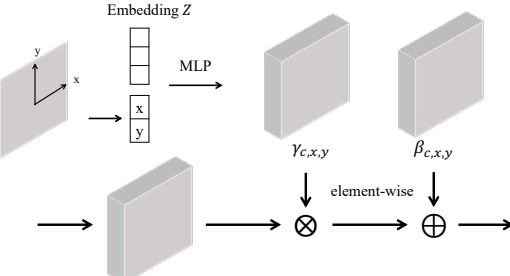

Figure 2: Spatially Adaptive LinEar Modulation (SALEM) Layer to incorporate contrastively learned embedding for dynamics forecasting

Intuitively, SALEM is designed to adaptively incorporate embedding knowledge into different spatial locations within the forecasting images. The normalization step helps keep the training process stable and consistent. The effectiveness of this approach will be demonstrated through quantitative results in the experiment section, and conceptual visualizations of SALEM's operation will be provided in Appendix C.4.

## 5 EXPERIMENTS

In this part, we evaluate our learning approach under multiple dynamical systems. For continuous time, we examine the spring-mass system and the Lotka-Volterra model (Lotka-Volterra, 1925), which serve as basic examples of linear and nonlinear systems, respectively. We start by examining the variation of system coefficients in a two-dimensional space, which allows for simpler visualization of how the learned representations correspond to these coefficients. Then we extend to the

original four-dimensional Lotka-Volterra model. For our evaluation of model performance in grid-based systems, we employ two types of models: incompressible flow systems and reaction-diffusion models. In both cases, we experiment with systems that have different numbers of varying coefficients. For dataset details, please refer to Appendix A.

## 5.1 Vector-based Systems

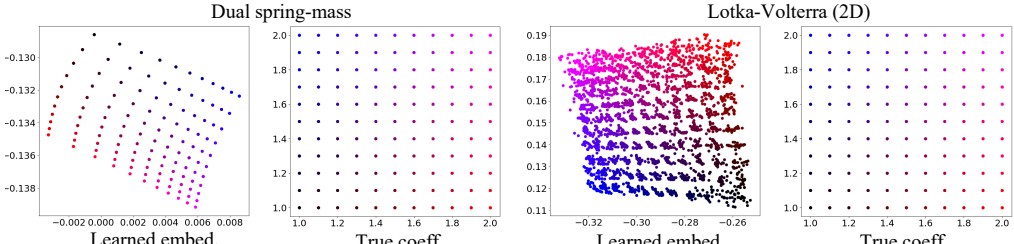

Figure 3: System coefficients (2nd and 4th picture) vs learned trajectory embedding (1st and 3rd picture) for dual spring-mass and Lotka-Volterra (2D) system. The x and y axes are learned embedding/true coefficients. For dual-spring mass plots, the color gradients represent the two spring rates. For Lotka-Volterra plots, the color gradient represent $\alpha$ and $\beta$ (defined in Appendix A).

In Figure 3, we depict the learned embeddings alongside the actual coefficients for two systems. The graph's axes show the embedding values learned by the model and the true system parameters in two dimensions. The color gradients always represent the true system parameters. For each system, we analyze 121 distinct coefficient sets, chosen with evenly spaced intervals in each dimension, totaling 11 per dimension. For every coefficient set, we initiate five distinct trajectories. In the case of the true system trajectories, since all five share identical coefficients, they overlap and only one trajectory for each set is visible.

In the dual spring-mass system case, the learned embedding is a rotated and scaled version of the true coefficients. This is acceptable since the embedding's form is unconstrained and the values are relative. Groups of five trajectories consistently converge due to the system's linearity, and the local linear extractor $\hat{A}_k$ is expected to directly represent the corresponding linear matrix. The Lotka-Volterra system behaves differently, with less alignment among the sets of five trajectories due to its inherent non-linearity. The variable nature of $\hat{A}_k$ in this context makes it challenging to extract uniform information across trajectories. Despite these complexities, the learned embedding still exhibits a roughly rotated shape, suggesting a two-dimensional variation that loosely correlates with the system's dynamical properties.

It is crucial to note that the model training was conducted in an unsupervised manner, without the incorporation of any prior information. Even under these conditions, the training process effectively ranks the different coefficients. This demonstrates the model's capability to discern and organize the coefficients in a meaningful way, despite the absence of explicit guidance or predefined knowledge.

After acquiring the embedding through contrastive learning, we apply this embedding for forecasting in dynamical systems. As detailed in Section 4.3 and illustrated in Figure 1, for forecasting purposes, we employ the observed trajectory to infer the embedding. This inferred embedding is then utilized to approximate the equations of the dynamical system. To streamline the process and avoid these complexities, we employ a simple MLP architecture in our method to evaluate the performance gain from contrastive learning. As a comparison baseline, the standard training method uses a similar MLP network with only system states $x$ as the input and performs supervised learning on all the given data. We present the forecasting results in Table 1. Our method consistently outperforms the baseline method across all the settings.

We delay the ablation studies and further experiments of the vector-based system to Section 6 and Appendix C.2 for readers' interest. To notice, we do not compare our model to other prior works (Yin et al., 2021; Kirchmeyer et al., 2022; Blanke & Lelarge, 2023) due to problem setting difference. To the best of our knowledge, all prior works in this area necessitate knowing the coefficients of the

Table 1: Simulation Error(MSE) for the Vector-based Systems

| Models | Spring-Mass | LV (2D) | LV (4D) |
|---|---|---|---|
| Standard training | 4.86e-4 $\pm$ 0.60e-4 | 4.40e-2 $\pm$ 0.44e-2 | 12.6e-2 $\pm$ 0.98e-2 |
| Our methods | **2.58e-4 $\pm$ 0.79e-4** | **1.31e-2$\pm$ 0.88e-2** | **8.31e-2$\pm$ 2.20e-2** |

system or require extra fine-tuning to adapt the model to the new systems, both will not fit into the above experiment setup. Our approach stands out from others as we deduce the system coefficients directly from observations, eliminating the need for model adaptation or few-shot learning when making predictions in new environments. Previous research can be considered as parallel to our forecasting stage, assuming the system coefficients are already known or learned from few-shot adaptation.

## 5.2 GRID-BASED PDE SYSTEMS

We experiment on two grid-based systems with three different setups for each: 1. Incompressible fluid flows: 1.1 varying buoyancy, 1.2 varying supply rate, 1.3 varying both buoyancy and supply rate, 2. Gray-Scott reaction diffusion system of two chemical species: 2.1 varying the feed rate 2.2 varying the killing rate 2.3 varying both feed and killing rate. The setup for the first experiment (1.1) is from Wang et al. (2022). Both datasets are synthetically generated by PhiFlow (Holl et al., 2020), and we leave the details to Appendix A.

We compare our learning method with 3 baselines: the standard ResNet, DyAN (Wang et al., 2022) and ResNet+FiLM module Perez et al. (2018). To notice, the DyAN method assumes prior knowledge of the system coefficients (e.g. pre-calculated vorticity) while our method does not require it. ResNet+FiLM is used for the ablation study of SALEM module. We present the results in Table 2. The three meta-learning methods outperform the standard ResNet by a large margin. Among the three, our SALEM method achieves the best performance. For certain experiments, DyAN and FiLM fail the training process and produces NaN during the prediction, while SALEM provides more stable training in practice. For visualization of the learned embedding and conceptual effectiveness of the SALEM layer, please refer to the results in Appendix C.3 and Appendix C.4.

Table 2: Simulation Error(MSE) for the Incompressible Fluid Systems[3]

| Models | buoyancy | supply rate | buoyancy & supply rate |
|---|---|---|---|
| Standard ResNet | 15.8e-2 $\pm$ 1.12e-2 | 4.66e-2 $\pm$ 0.13e-2 | 4.05e-1 $\pm$ 1.66e-1 |
| DyAN | 9.51e-2 $\pm$ 2.21e-2 | 3.86e-2 $\pm$ 0.88e-2 | N/A |
| ResNet + FiLM | 9.90e-2 $\pm$ 1.48e-2 | 4.1e-2$\pm$ 0.08e-2 | NaN |
| ResNet + SALEM (ours) | **9.06e-2 $\pm$ 2.77e-2** | **3.12e-2 $\pm$ 0.84e-2** | **1.60e-1 $\pm$ 0.73e-2** |

Table 3: Simulation Error(MSE) for the Gray-Scott Systems

| Models | feed rate | kill rate | feed & kill rate |
|---|---|---|---|
| Standard ResNet | 3.49e-3 $\pm$ 0.56e-4 | 1.53e-3 $\pm$ 0.44e-3 | 4.62e-3 $\pm$ 0.29e-3 |
| DyAN | 31.1e-3 $\pm$ 9.62e-3 | NaN | N/A |
| ResNet + FiLM | 4.15e-3 $\pm$ 1.32e-4 | 1.38e-3 $\pm$ 0.48e-4 | 4.56e-3 $\pm$ 1.82e-3 |
| ResNet + SALEM (ours) | **2.28e-3 $\pm$ 1.87e-4** | **1.08e-3 $\pm$ 0.36e-3** | **3.81e-3 $\pm$ 2.22e-3** |

## 6 DISCUSSIONS

### 6.1 ABLATION STUDIES

To evaluate the contribuiton from different techniques in Section 4, we also provide ablation studies in Table 4. In our baseline setup, the covariance regularize coefficient $\lambda$ is set to 0.5 in Equation (4). To assess its sensitivity, we adjusted this value to 0.2 and found the proposed method is not sensitive to this hyper-parameter. However, completely removing this term from the loss function can lead to

---

[3]"NaN" indicates training failed (loss goes to NaN) or prediction became unstable (error goes to NaN), "N/A" indicates not applicable.

a marked increase in error. If conventional contrastive loss functions are adopted, such as Info-NCE or Triplet loss, they result in a significant increase in prediction error. We also tested standard LSTM without local feature extractor for trajectory mapping, the performance is slightly worse, therefore, we choose the local feature extractor as standard setup.

Table 4: Ablation Study for the Vector-based Systems

| Loss | Covariance Reg. | Local feature extractor | Dual Spring-Mass | LV (2D) | LV (4D) |
|---|---|---|---|---|---|
| ESR | $\lambda = 0.5$ | ✓ | 2.58e-4 ± 0.79e-4 | **1.31e-2± 0.88e-2** | 8.31e-2± 2.20e-2 |
| ESR | $\lambda = 0.2$ | ✓ | **2.54e-4± 0.22e-4** | 2.31e-2± 0.44e-2 | 8.61e-2± 1.57e-2 |
| ESR | ✗ | ✓ | 4.59e-4 ± 0.63e-4 | 2.89e-2± 0.69e-2 | 10.7e-2± 2.37e-2 |
| ESR | $\lambda = 0.5$ | ✗ | 4.92e-4 ± 0.69e-4 | 1.68e-2± 0.46e-2 | **7.73e-2± 1.37e-2** |
| Info-NCE | ✗ | ✓ | 2.75e-4± 0.47e-4 | 3.89e-2± 0.22e-2 | 22.4e-2± 5.39e-2 |
| Triplet | ✗ | ✓ | 6.01e-4± 1.63e-4 | 4.23e-2± 0.88e-2 | 12.2e-2± 1.96e-2 |

## 6.2 COMPARISON WITH PRIOR WORKS

Recently, the field of meta-learning for dynamical systems has begun to gain attention, with a small but growing number of studies endeavoring to tackle this complex issue. Our approach, from the formulation of the problem to the learning techniques used, significantly distinguishes from previous methods.

Our method assumes no provided label information regarding the system coefficients, thus not requiring the knowledge of new system coefficients for either supervised system embedding learning or in predicting a new system. This is the major difference between our method and prior works. Wang et al. (2022) needs the system coefficients for weak supervised learning. Yin et al. (2021); Kirchmeyer et al. (2022) assume the linear decomposition of the shared dynamics and environment-specific dynamics, where the latter requires unique model parameters fitted separately for each environment. These strong conditions facilitate the learning process as they acquires knowledge about knowledge for sub-systems. However, in many instances, especially in forecasting scenarios, this kind of information is often not available.

Once the embedding is learned, our approach is compatible with other meta-learning method Yin et al. (2021); Kirchmeyer et al. (2022) to quickly adapt to new vector-based systems. For the grid-based systems, similar to Wang et al. (2022), we generalize the environment-specific embedding from the input and utilize this mapping for forecasting tasks of new systems, while our approach does not require explicit knowledge for this embedding learning.

## 7 CONCLUSION AND LIMITATIONS

In this paper, we present a novel meta-learning framework for dynamical systems, leveraging contrastive learning coupled with a tailored feature extractor and a custom loss function. This architecture facilitates the extraction of embeddings that capture the underlying physical parameters of the system. We further enhance this model with an innovatively designed forecasting module, which utilizes the model embeddings for more accurate prediction. Our approach not only achieves superior forecasting accuracy but also enhances model interpretability and physical relevance.

As we aim to build a contrastive learning framework for general dynamical systems, we acknowledge several limitations in the current study. Firstly, the complexity of the local linear extractor grows quadratically with the system dimension, and we have yet explored its application for high-dimensional vector-based systems. Furthermore, the embedding space utilized in this paper is also of a low dimensionality. The potential of high-dimensional embedding spaces remains unexplored, where we anticipate that the requisite data size could increase exponentially with the dimensionality of the embedding space. We hope our work can inspire future researchers on more universally applicable unsupervised learning methods that can handle high-dimensional continuous embedding space.

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

## A  DATASET DETAILS

### A.1  DUAL SPRING MASS SYSTEM

The system equation is formed as following:

$$m_1 \frac{d\dot{x}[1]}{dt} = k_2(x[2] - x[1]) - k_1 x[1]$$
$$m_2 \frac{d\dot{x}[x]}{dt} = -k_2(x[2] - x[1])$$
$$\frac{dx[1]}{dt} = \dot{x}[1]$$
$$\frac{dx[2]}{dt} = \dot{x}[2]$$

where the system state $[x[1], x[2], \dot{x}[1], \dot{x}[2]]$ represents the positions and velocities of two masses. The parameters $k_1, k_2, m_1, m_2$ correspond to the spring constants and the masses associated with these two objects. We standardize the mass constants by setting $m_1, m_2 = 1$ and vary the values of $k_1, k_2$ by sampling from $[1, 2]$.

### A.2  LOTKA-VOLTERRA SYSTEM

The Lotka-Volterra system Lotka-Volterra (1925) describes the dynamics of biological systems in which two species interact as a predator and a prey. The equations are shown in Equation (10) with four system coefficients: $\alpha, \beta, \gamma, \delta$. For visualizing the qualitative study of the embedding learning (LV-2D), we first fix $\gamma = 1, \delta = 1$, and sample $\alpha, \beta$ from $[1, 2]$. In the latter case of forecasting tasks (LV-4D), we sample all $\alpha, \beta, \delta, \gamma$ from $[1, 2]$.

$$\dot{x}[1] = \alpha x[1] - \beta x[1]x[2] \tag{10}$$
$$\dot{x}[2] = \delta x[1]x[2] - \gamma x[2]$$

For the above vector based systems, during training situations where coefficients vary across two dimensions, we generate samples from 64 distinct sets of dynamical systems (we provided data efficiency experiment in Table 5). For each system, we select 5 unique starting points and from each, produce 5 separate trajectories. These trajectories span a duration of 10 seconds, with data points collected at intervals of 0.1 seconds.

In the standard testing stage, we sample the same number of sets of coefficients from the same distribution (we also provides OOD testing experiments in Table 6). For each coefficient, we generate 5 trajectories and truncate them into sequences of 3.0 seconds, at intervals of 0.1 seconds. The initial 2.0 seconds of each trajectory are utilized to deduce the embeddings, following which we predict the system behavior for the subsequent 1.0 seconds and compare these forecasts to the actual data.

### A.3  INCOMPRESSIBLE FLUID WITH INLET FLOW AND BUOYANCY FORCE

This example was customized from a demo case of the PhiFlow Package [4] and was used in Wang et al. (2022) as a showcase example under varying buoyancy factors. The fluid dynamics were solved using a divergence-free solver to enforce the continuity equation, ensuring mass conservation. Advection of the fluid properties was handled using a semi-Lagrangian solver for numerical stability. The simulation setup included a square-shaped inlet flow with a specified area and flow rate, introducing fluid into the domain. Additionally, buoyancy forces were incorporated to capture the effects of density differences within the fluid. In our work, we vary the buoyancy factor and the inlet flow supply rate to generate a diverse dataset of fluid flow fields.

---

[4]https://github.com/tum-pbs/PhiFlow/tree/1.0.1

- Incompressible Flow with different buoyancy factor: We use the dataset introduced in Wang et al. (2022). The buoyancy factor is varied from 1 to 25 and 25 trajectories were generated. 20 trajectories/set of coefficients were randomly drawn for training and the remaining 5 is used for testing (this is same across all the grid based experiments). Each trajectory has 500 steps with 1 second interval.

- Incompressible Flow with different supply rate: In this example, we fixed the buoyancy factor to 5 and vary the initial supply rate of the fluid from 1 to 8. A total of 25 inflow rates are used to generate the data.

- Incompressible Flow with different buoyancy factor and supply rate: In this example, we vary the buoyancy factor from 1 to 5 and vary the initial supply rate of the fluid from 1 to 5. A total of 25 sets of coefficients are used to generate the data.

## A.4 GRAY-SCOTT REACTION DIFFUSION SYSTEM

$$\frac{\partial u}{\partial t} = D_u \nabla^2 u - uv^2 + f(1 - u)$$

$$\frac{\partial v}{\partial t} = D_v \nabla^2 v + uv^2 - (f + k)v$$

The Gray-Scott system is a type of reaction-diffusion model that describes the behavior of chemical reactions and diffusion processes, characterized by two interacting chemical species ($u$ and $v$ in the above equation). It is a specific form of the more general reaction-diffusion systems used to model various natural phenomena such as animal coat patterns, chemical reactions, and biological morphogenesis. $f$ is the feed rate of species $u$ and $k$ is the kill rate of species $v$.

- Reaction diffusion system with different feed rate: we vary feed rate ($[0.01, 0.1]$) only and generate 25 trajectories.

- Reaction diffusion system with different kill rate: we vary kill rate ($[0.06, 0.064]$) only and generate 25 trajectories.

- Reaction diffusion system with different feed rate and kill rate: we vary both feed rate ($[0.01, 0.1]$) and kill rate ($[0.06, 0.064]$) and generate 25 trajectories.

## B EXPERIMENT DETAILS

### B.1 COMPUTATIONAL PLATFORM

We perform all the experiments on a single RTX4090 GPU. For the quantitative results, we conduct each experiment over 5 random seeds and report the average.

### B.2 NEURAL NETWORK STRUCTURES

#### B.2.1 VECTOR-BASED SYSTEMS

For the embedding networks, we use a standard three layer LSTM with hidden varible size of 100. With the last hidden value of LSTM as input, we then use a fully connected neural network with 2 layers to get the final output. Each hidden layer has 100 neurons.

Regarding the dimension of the embedding, it is determined based on the count of coefficients that vary. In the case of the two-spring mass system, this dimension is set to 2. For the Lotka-Volterra system, the embedding dimension is either 2 or 4, depending on the number of coefficients that are variable.

For the forecasting neural network, we simply use a fully connected neural network with 2 hidden layers. Each of the hidden layer has 100 neurons. We combine the system state and trajectory embedding directly by concatenation to form the input, so the input layer size is the sum of these two dimensions.

For dynamical system forecasting, we generate trajectories span a duration of 2 seconds (20 timestamps) as input for the model. We predict the future trajectory for 1 second (10 timestamps) and evaluate it with the ground truth data.

### B.2.2 GRID-BASED SYSTEMS

In the embedding stage, we use a standard CNN (with channel sizes of 64-128-256-512 and kernel size of 3 across all the layers) followed by a two layer MLP to mapping the grid state under each time step to a hidden vector of size 5. Then we use a standard three layer LSTM + MLP as above to map the sequential input to an embedding. During the contrastive training, we use the embedding size of 1. During the forecasting stage, we drop the last layer and use the output of the second last layer with a dimension of 5 to incorporate the system knowledge to forecasting models.

For the forecasting model, the backbone structure we use is a 18-layer ResNet with kernel size of 3 across all the layers. The same configuration is used across all the model variants. With the FiLM model, in each convolutional layer, we use linear layers to map the embedding vector to the affine coefficients (with the same size as the output channels). In our SALEM model, we use a two layer MLP to map the concatenation of embedding vector and pixel coordinates to the affine coefficients (with the same size as the output channels).

### B.3 TRAINING DETAILS

Across all the experiments, we use a Adam optimizer with a learning rate of 0.001.

In the embedding learning stage, each trajectory is spliced into 5 truncated trajectories to feed into the contrastive training. For vector-based systems, we use a batch size of 200 and training the model for 2000 epochs. For grid-based systems, we use a batch size of 16 and training the model for 2000 epochs.

In the forecasting training stage, for vector-based systems, we use a batch size of 200 and training the model for 1000 epochs. For grid-based systems, we use a batch size of 16 and training the model for 50 epochs. In the training stage, we forward the discrete system for 3 time steps, and compare with the ground truth data. In the testing stage, we evaluate the model performance by forecasting 20 time steps.

## C MORE EXPERIMENT RESULTS

### C.1 CONTRASTIVE LEARNING LOSS FUNCTION CHOICE AND ABLATION STUDIES

In classical applications of contrastive learning such as computer vision, the target of learning process is to form discrete clusters in the embedding space, where each cluster represents a specific category or class. This learning objective is a direct result of the inherent categorization of the data and is explicitly influenced by the design of contrastive loss functions (Oord et al., 2018a; Chen et al., 2020a;b). These loss functions, often in conjunction with softmax or exponential functions, work by assigning probabilities to pairs of samples, quantifying how likely it is that a pair of embeddings belongs to the same class. This probabilistic approach allows the model to effectively determine how closely an anchor point (a reference data point) is related to its positive pairs, compared to various negative pairs, facilitating the formation of distinct, class-based clusters in the embedding space.

In contrast, when applied to dynamical systems, contrastive learning results in a more continuous distribution of embeddings rather than distinct clusters. This difference stems from the continuous nature of dynamical systems, where the coefficients of the sub-systems are smoothly interconnected.

To assess the effectiveness of various embedding learning modules, we systematically replaced each module and analyzed the resultant embeddings. Figure 4 presents the outcomes of these ablation studies. In the top-left figure, the absence of covariance loss leads to embeddings that tend to align along a line, suggesting a linear correlation between latent dimensions and resulting in dimension collapse due to this linear dependency.

The mid-left figure illustrates the results using the standard square ratio loss from Zhang et al. (2023), where a similar linear arrangement of embeddings is observed. However, the rescaling of

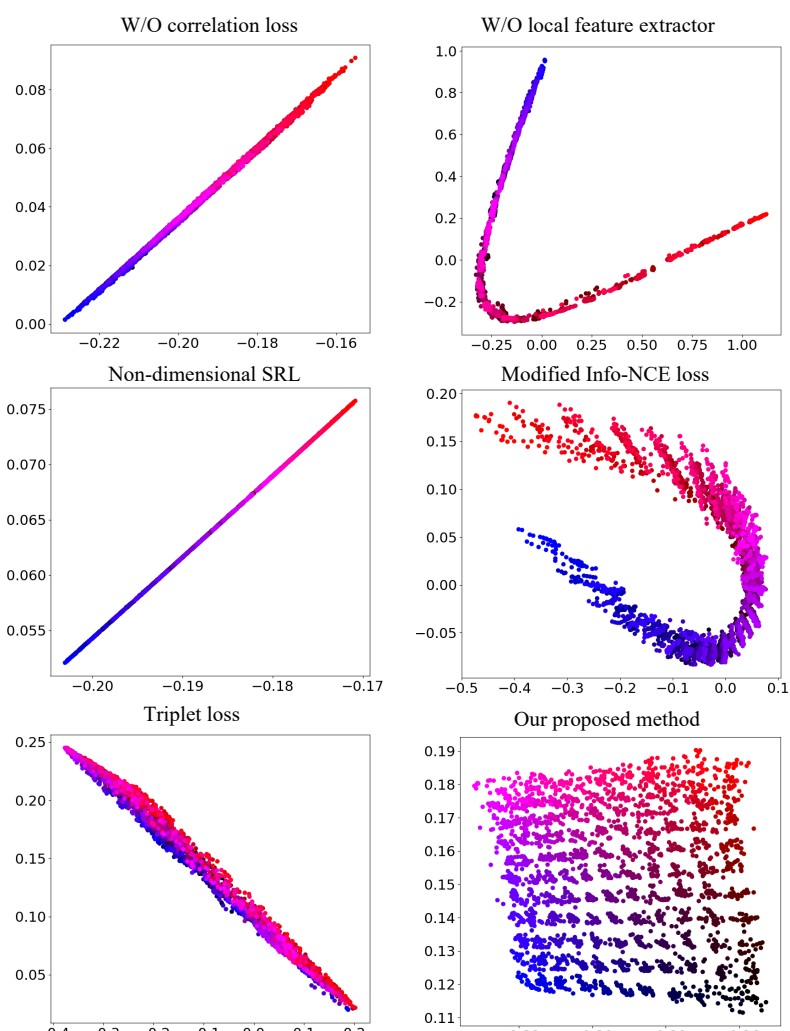

Figure 4: Ablation study on embedding learning: typical learned embeddings

axes leads to reduced variation along the horizontal axis compared to the vertical one, implying that only one embedding dimension is effectively conveying information, which again culminates in dimensional collapse.

Further adaptations were made to the InfoNCE loss and Triplet loss to include multiple positive pairs, with findings illustrated in the bottom-right figure. As in Wang & Isola (2020), the embeddings tend to uniformly distribute over a hypersphere, although some areas of the distribution still show signs of dimensional reduction. The Trilplet loss cannot learn an organized embedding space.

In the top right figure, we used trajectory states directly as inputs to the LSTM, bypassing the local linear least square feature extractor as delineated in Equation (5). This approach demonstrated that standard LSTMs typically struggle to extract robust embedding information. In the local linear least square feature extractor, the goal of deriving the system matrix is to collect data for deducing system coefficients at a local level. It could be debated that this local linearity does not necessarily equate to the non-linear system, or that equation Equation (5) might be ill-conditioned for solving. Nevertheless, by utilizing the estimated $\hat{A}_k$ from various truncated sequences, this approach will collect insights from diverse samples and ultimately extract the valuable information.

## C.2 EXTRA EXPERIMENTS FOR VECTOR-BASED DYNAMICAL SYSTEMS

We conducted additional experiments to comprehensively assess the proposed method's performance across various conditions.

Table 5 shows the model data efficiency with different numbers of trajectories used for training. The main paper's standard experiments use the middle column's data (64 trajectories for 2D systems, 256 for 4D systems). We explore the effects of both reducing and increasing this data. Across all scenarios, our suggested approach consistently outperforms conventional methods.

Table 5: Experiment over Different Trajectory Number for the Vector-based Systems

| Method | Data | # of training traj. | | |
|---|---|---|---|---|
| | | 25 | 64 | 121 |
| Standard training | Spring-Mass | 4.92e-4 $\pm$ 0.63e-4 | 4.86e-4 $\pm$ 0.60e-4 | 4.82e-4 $\pm$ 0.31e-4 |
| Our methods | | **4.62e-4** $\pm$ 0.58e-4 | **2.58e-4 $\pm$ 0.79e-4** | **6.92e-5** $\pm$ **2.89e-5** |
| | | # of training traj. | | |
| | | 25 | 64 | 121 |
| Standard training | LV (2D) | 4.69e-2 $\pm$ 0.93e-2 | 4.40e-2 $\pm$ 0.44e-2 | 4.55e-2 $\pm$ 0.59e-2 |
| Our methods | | **1.63e-2 $\pm$ 0.84e-2** | **1.31e-2** $\pm$ **0.88e-2** | **1.74e-2** $\pm$ **1.20e-2** |
| | | # of training traj. | | |
| | | 81 | 256 | 625 |
| Standard training | LV (4D) | 17.0e-2 $\pm$ 3.72e-2 | 12.6e-2 $\pm$ 0.98e-2 | 13.5e-2 $\pm$ 0.57e-2 |
| Our methods | | **7.85e-2 $\pm$ 1.23e-2** | **8.31e-2** $\pm$ **2.20e-2** | **8.69e-2** $\pm$ **1.47e-2** |

Table 6 shows the model performance in out of distribution data. Across all three experiments, the testing coefficients were sampled from $[0.8, 1]$ instead of $[1, 2]$ from the training distribution and standard testing distribution in the main paper. Our proposed meta learning methods still outperforms the standard methods.

Table 6: Simulation Error(MSE) of OOD testing data for the Vector-based Systems

| Models | Spring-Mass | LV (2D) | LV (4D) |
|---|---|---|---|
| Standard training | 3.37e-3 $\pm$ 0.28e-3 | 3.97e-2 $\pm$ 0.57e-2 | 10.1e-2 $\pm$ 1.27e-2 |
| Our methods | **1.01e-3 $\pm$ 1.47e-3** | **2.12e-2 $\pm$ 0.89e-2** | **8.03e-2 $\pm$ 3.22e-2** |

In Table 7, we evaluated our proposed methods against a benchmark that uses known coefficients as additional input for the forecasting model. While this comparison is not fair, as it provides both training and testing data with known information, our goal is to investigate how well our contrastive learning approach can identify differences between systems. As expected, the experiments using known coefficients outperformed the other methods. However, our proposed approach showed promising results, coming quite close to the benchmark, particularly in the initial two experiments. This suggests that our method has capability for effectively extracting inter-system variations without relying on prior knowledge of coefficients for the low dimensional systems. It is also be valuable for complex scenarios, while it might not capture all nuances in high-dimensional contexts.

Table 7: Experiment with Known Coefficients for the Vector-based Systems

| Methods | Spring-Mass | LV (2D) | LV (4D) |
|---|---|---|---|
| Standard training | 4.86e-4 $\pm$ 0.60e-4 | 4.55e-2 $\pm$ 0.59e-2 | 12.6e-2 $\pm$ 0.98e-2 |
| Our methods | 2.58e-4 $\pm$ 0.79e-4 | 1.31e-2 $\pm$ 0.88e-2 | 8.31e-2 $\pm$ 2.20e-2 |
| Known coeffs. | **2.39e-4 $\pm$ 1.17e-4** | **0.85e-2 $\pm$ 1.01e-2** | **2.12e-2 $\pm$ 0.81e-2** |

## C.3 LEARNED EMBEDDING FOR GRID-BASED DYNAMICAL SYSTEMS

We show the typical learned embedding of the grid-based system experiments in Figure 5 and Figure 6. The x-axis is the varying coefficients and the y-axis shows the the learned embedding. Each dataset contains 25 trajectories, 20 of which are used for training and the other 5 used for evaluation.

Each trajectory is divided into five segments, with each segment corresponding to one of the five overlapping points located under each x-axis value in the figure. Despite some instances where the points do not entirely overlap or the trend fluctuates locally, there is generally a monotonic relationship between the learned embedding and the actual coefficients. This information will be utilized for forecasting in dynamical systems. It is important to note that the order of the embeddings is learned automatically without any supervised learning techniques.

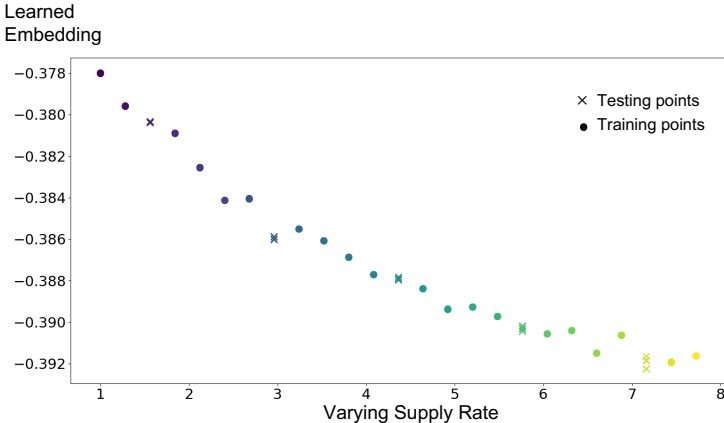

Figure 5: Learned Embedding vs System Coefficient: Incompressible Fluid with Varying Supply Rate

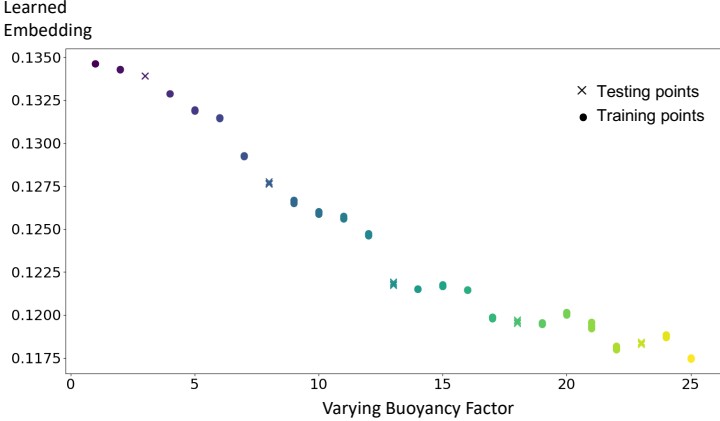

Figure 6: Learned Embedding vs System Coefficient: Incompressible Fluid with Varying Buoyancy Factor

### C.4 VISUALIZATION OF SALEM LINEAR MODULATION

To conceptually visualize the effect of the SALEM layer, we select the early stage (20 step after the initial step) of 20 trajectories with varying coefficients. We then compute the standard deviation across these trajectories and display their mean in the upper row of Figure 7. Additionally, during the forward pass of the neural network, we calculate the average absolute value of $\gamma$ (defined in Equation (8)) across the channels in the first SALEM layer and visualize them in the bottom row. The grayscale intensity in the images thus indicates the relative value of the standard deviation and average absolute $\gamma$ across different spatial locations. The left column is for the varying buoyancy factor case, and the right column is for the varying supply rate case with incompressible fluids.

When the buoyancy varies, because the supply rate is identical, the trajectory variance is primarily evident in the upper portion of the image, where buoyancy is the dominant factor. This observation

is consistent with the average magnitude of $\gamma$, which is larger at the top of the image. Conceptually, this suggests that the linear modulation places more emphasis on incorporating knowledge from the embedding into the upper region of the simulation domain, where the effects of buoyancy are more pronounced. The inlet flow supply is located in the central region of the simulation domain. As a result, when the supply rate varies in the initial stages, the data exhibits variation in the middle section of the field. This variation is reflected in the corresponding mean value of $\gamma$, which is also higher in the central area. However, due to the simplicity of the MLP used for linear modulation, the model is unable to capture the precise shape of the supply rate variation. Instead, it can only roughly capture the variation in the middle portion of the image.

Figure 7: Trajectory-wise data variation and mean $|\gamma|$ in the two grid-based systems (grayscale indicates the relative value in each picture)

