# OpenReview forum: "Contrastive Meta Learning for Dynamical Systems"
_ICLR.cc/2025/Conference — Submitted to ICLR 2025_

### Official Review · Reviewer_Fmy9 · 2024-11-01

**Soundness:** 2
**Presentation:** 3
**Contribution:** 3
**Rating:** 5
**Confidence:** 4

**Summary:**

This paper aims to solve the following problem: if we have some time-series data from a dynamical systems at several different parameters, can we forecast the dynamics at a different parameter not in the training set. The authors achieve this by first identifying the unknown parameter from a trajectory observation using contrastive learning (embedding the trajectory to infer the parameters), they then use the inferred parameter to make forecasts.

**Strengths:**

Forecasting dynamical systems at previously unseen parameters is an important and challenging problem that has resisted many attempts from the nonlinear dynamics and scientific machine learning communities. This paper presents an interesting idea using contrastive learning that could prove useful towards solving this longstanding problem.

**Weaknesses:**

With the current experiments, I am not fully convinced that the proposed approach would work as well as claimed in complex and real-world situations. I will explain more with my questions below.

**Questions:**

1. Figure 1 suggests that dynamical systems with the same parameter but different initial conditions have similar dynamics, while those with different parameters show distinct behaviors. However, many dynamical systems are multistable, meaning exactly the same system when starting from different initial conditions can exhibit very different dynamics (i.e., reaching different attractors). How does the proposed method address this challenge?

2. Concerning Eqs. (2) to (4), they address dimensional collapse due to linear correlations, but what about other forms of more nonlinear relation between different dimensions (which can also lead to dimensional collapse)?

3. Eq. (5) relies on estimating time derivatives from data, which is known to be sensitive to noise. This raises questions about how robust the proposed method is.

4. The experiments were performed on a few simple systems such as the spring-mass system and the Lotka-Volterra model. What about other more complex systems? For example, chaotic ones or higher-dimensional ones? Or ones whose dynamics change dramatically with parameters (e.g., going through bifurcations).

5. Moreover, the model space currently only contain one class of equations (e.g., the Lotka-Volterra model). What if it spans several classes of qualitatively different systems? Can the proposed method handle this more general situation?

6. Data from how many different parameter sets are required for effective training?

---

> ### Author Response · Authors · 2024-11-19
> **Rebuttal reply to Reviewer Fmy9**
>
> Dear reviewer:
>
> We thank you for the insightful questions, let us reply them in the following:
>
> $\bullet$ Multistable dynamics:
>
> we thank you for the observation, Figure 1 is just an intuitive explanation of the proposed method, and does not indicate that the system has to converge to a single attractor or behave periodically. In the contrastive learning part, we are learning a mapping from a truncation of trajectory (usually 5-10 steps) to the embedding which could represent the system coefficients. Taking an extreme but basic system identification problem with linear system ($dx/dt=Ax+b$), if we want to use a linear model to approximate it, we only need some time steps to characterize A and b, and we don’t need to assume the trajectories are converging or close to each other. Especially when we use the local linear feature extractor, the time evolution of the state matters much more than the absolute coordinates.
>
> $\bullet$ Dimensional collapse for nonlinear correlation between different dimensions:
>
> The covariance loss is inspired by Ref [1] below. Unfortunately, we cannot prevent this from happening by linear covariance regularizer as it cannot capture the nonlinear correlation. However, such a scenario is rare. Because the different dimensions are all from the last fully connected neural network, linear correlations are much more likely to happen. In practice, we never meet nonlinear correlations (e.g. the 2D embedding distribution shrinked into a curve/circle). The only scenario when the dimensional collapse happens shows a linear relationship in the embedding space.
>
> Ref 1: VICReg: Variance-Invariance-Covariance Regularization for Self-Supervised Learning, ICLR 2022
>
> $\bullet$ Equation 5:
>
> Yes, we agree with you that the noise might affect the derivative approximation for local feature extractor. In our method, this feature extract functions as a quick and rough pre-processing of the data. Since we use multiple steps to approximate each local $A$ and a trajectory of $A$ is flattened as LSTM input, we do not require each approximation to be accurate.
>
> $\bullet$ More complex systems:
>
> We provide high-dimensional system experiments in the grid-based system study (Section 5.2 where the system has an image-like input). Since this research serves primarily as a proof of concept in this new field, we do not include experiments with chaotic or bifurcation systems. Nevertheless, we anticipate that meta-learning's performance would likely degrade when dealing with chaotic systems, particularly when test coefficients produce chaotic behaviors different from those in the training data. This is because meta-learning aims to extract and generalize common patterns across different environments. In chaotic systems, even slight deviations from the training distribution can lead to dramatically different behaviors, which may be beyond the meta-learning algorithm's ability to capture. Again, our experiments are designed to demonstrate the feasibility of our proposed algorithm. Given that system identification of chaotic systems remains challenging even for traditional methods, we choose to maintain a focused scope rather than expand into more complex chaotic scenarios.
>
> $\bullet$ qualitatively different systems:
>
> Thank you for the suggestion for bringing up a new direction. If different types of equations are introduced, then the “environment” space needs to be redefined as a new functional space. The proposed method in this could be a starting point/baseline for this new problem. Suppose we have a few different equation classes, I would suggest using probabilistic focused contrastive learning losses (e.g. Info-NCE) to train it, as it fits better for classification problems than regression on continuous space. However, it still would be challenging for the meta learning method to perform well especially on an unseen type of equations.
>
> $\bullet$ How many parameter sets are required for effective training?
>
> In appendix A, we provide all the parameter set/experiment details for standard experiment in the main paper. We also provide extra experiments for the performance gain with the proposed method over the standard training method under different numbers of coefficient sets in table 5 in Appendix C. The proposed method achieves better accuracy than standard experiment across all the setups. While “effective training” is a relatively ambiguous word, we believe as long as the systems exhibit shared characteristics and distinct variations, the meta learning method can hold an advantage because it utilizes additional information from the datasets.

---

> > ### Comment · Reviewer_Fmy9 · 2024-11-25
> >
> > Thank you for the clarification.

---

### Official Review · Reviewer_EvLj · 2024-11-02

**Soundness:** 3
**Presentation:** 3
**Contribution:** 3
**Rating:** 6
**Confidence:** 3

**Summary:**

This paper studied forecasting in dynamic systems in continuous and discrete setup via zero-shot meta learning. The proposed framework has three steps:

1- employing contrastive learning via a modified loss Element-wise Square Ratio Loss (ESR) to derive embeddings in both intra- and inter-truncated systems (meta step 1)

2- applying this embedding model to the initial segment of the data to obtain embeddings (meta step 2)

3- utilizing the final embeddings in a prediction module for forecasting

Some novel aspects of the work are proposing a Local Linear Least Square feature extractor for vector-based systems and a Spatial Adaptive LinEar Modulation (SALEM) for discrete systems.
Authors conducted various experiments and ablation studies and demonstrated the utility of their framework.

**Strengths:**

- The paper is well-written and well-organized.
- The paper proposed the first method of zero-shot meta learning for dynamic systems
- There are quite a few novel aspects as well as the main contribution

**Weaknesses:**

The setup is highly relevant to domain generalization/out-of-distribution generalization task and there are methods of meta-learning for that tasks such as [1]. There is no indication of this in the paper/baselines. Only in the appendix one experiment is provided that compares the proposed method to a standard training setup.

Although authors studied the triplet loss and the Info-NCE loss, I think another relevant loss for this particular application is the Histogram loss [2] because of its probabilistic nature - it is defined on the similarity distributions of positive and negative pairs where distributions are estimated based on histograms. Authors should either justify that it is not relevant or include it in the experiments.

I can understand authors viewpoint by not comparing their method to baselines in the vector-based experiments, but I still believe including those results will increase the impact of the paper. If the proposed method is still better than the baselines even though baselines require additional information, that would increase the contribution of the paper. On the other hand, if the baselines outperform the proposed method then I’d like to see a few-shot version of the method.

There is no indication of the code and releasing it. I strongly encourage authors to either include a statement about code release plans in their paper, or provide a link to a code repository if it's already available.


There is no instruction on how to tune lambda hyper-parameter. Why 0.5 is the maximum value for it? it would be great to provide details on how they selected the lambda value, and if they explored a range of values other than 0, 0.2, or 0.5.

If Table 3 and 4 also serve as ablation for the discrete systems, then ResNet+SPADE is missing and should be added because it would represents methods that are based on spatial information compared to the proposed method that utilizes both channel embeddings and spatial information or authors need to justify why it is not necessary.

[1] Da Li et al. Learning to Generalize: Meta-Learning for Domain Generalization. AAAI 2017
[2] Evgeniya Ustinova and Victor Lempitsky Learning Deep Embeddings with Histogram Loss. Neurips 2016

**Questions:**

- What would be the impact of deeper backbones for ResNet and LSTM in the driving application?

- How does the performance change by expanding the input window and/or increasing the prediction horizon in both grid-based and vector-based systems?

- There is no indication of any regularization technique in the reproducibility details, was there any? Also, meta learning methods are often hard to train. I think providing learning curves and training dynamics would increase the reproducibility and debugging of the work for future users.

- What makes these two different losses (ESR and Cov) combinable into one loss e.g., based on their nature and what they represent and the scale? it would be great if authors can provide more theoretical justification for combining these losses or reasoning and explain the rationale behind penalizing perfect correlation in the covariance loss.

- Meta learning models are usually time consuming and authors eluded the computational complexity of the feature extractor a bit in the conclusion, how much do you increase time/space complexity compared to standard training in the vector-based system and studied baselines in the grid-based system?

I'd be happy to increase my score if authors respond to my questions here and points raised in the weakness, particularly:

1- Potentially missing experiments (Feel free to include them or provide your reasoning why they are not necessary) e.g., missing ablation, missing baselines in the vector-based case, or other metric learning losses.

2- Questions and concerns related to the methodology e.g., combining losses or complexity

3- Questions and concerns related to reproducibility

Update: I would like to thank authors for their engagement during the rebuttal period. After reading their response and other reviewers' comments I will increase my confidence and remain in favor of acceptance.

---

> ### Author Response · Authors · 2024-11-19
> **Rebuttal reply to Reviewer EvLj part [1/2]**
>
> We thank you for the thorough reading of the paper and overall positive comment.
>
> $\bullet$ Baselines:
>
> As meta learning for dynamical systems is a new area, there is limited prior work. For the grid based PDE problem, we do compare to and slightly outperform the existing baselines such as DyAN ref[2] in our table 2 and 3, despite their encoder being learned in the supervised way instead of our unsupervised learning. This echoes with your comments “If the proposed method is still better than the baselines even though baselines require additional information, that would increase the contribution of the paper.”
>
> For the vector based systems, there are existing works such as ref[3][4]. Similar to the paper you referred to for the more general meta learning setup (Ref [1]), they all require additional data for the fine-tune process when a new testing environment is introduced. In our problem setup, we do not have additional data for the new environments, therefore all the existing methods are not applicable and so we can only compare with standard training. To fairly compare with these baselines, we need to dramatically modify the training pipeline and the data generation process, and this opens a new discussion of how much data is needed for few-shot learning (if limited data is added, then our proposed method will perform better; otherwise the fine-tune methods will do certainly do better). Due to limited rebuttal time, we are not able to perform this level of code change and incorporate other people’s work into our repo. We hope you can understand this, as we are the first to propose unsupervised pre-training + zero-shot learning for meta-learning dynamical systems.
>
> [1] Learning to Generalize: Meta-Learning for Domain Generalization. AAAI 2017
>
> [2] Meta-learning dynamics forecasting using task inference, NeurIPS 2022
>
> [3] Learning dynamical systems that generalize across environments. Advances in Neural Information Processing Systems, NeurIPS 2021
>
> [4] Generalizing to new physical systems via context-informed dynamics model, PMLR 2022
>
>
> $\bullet$ Histogram loss:
>
> Thank you for your suggestion, we managed to test this loss function as baseline/ablation studies. In general, the performance of histogram loss is similar to triplet loss and worse than our proposed method. Here are the results for Histogram loss compared with standard training and our proposed method and we will update to our table 4 and cite the paper in the next version.
>
> |**Method**|**Dual spring mass** |**LV-2D**|**LV-4D**|
> |:--|:--|:--|:--|
> |Ours|**2.58e-4±0.79e-4**|**1.31e-2±0.88e-2**|**8.31e-2±2.20e-2**|
> |Triplet|6.01e-4±1.63e-4|4.23e-2±0.88e-2|12.2e-2±1.96e-2|
> |Histogram|11.63e-4±2.75e-4|4.65e-2± 0.84e-2|9.65e-2±1.71e-2|
>
> $\bullet$ Code release:
>
> Yes we will surely release the code once the paper is accepted.
>
> $\bullet$ lambda hyper-parameter:
>
> We do provide the lambda tuning experiment in table 4 with lambda = 0,0.2,0.5. The experiments reveal a wide tolerance window for this tuning since 0.2/0.5 does not change the result dramatically and both of them can output an embedding mapping similar to Figure 3. The purpose of this regularizer is to prevent collapse of embedding dimensions and 0.5 is strong enough for this purpose. Further increasing the number will not improve it any more. Therefore we choose 0.5.

---

> > ### Author Response · Authors · 2024-11-19
> > **Rebuttal reply to Reviewer EvLj part [2/2]**
> >
> > $\bullet$ Additional ablation study for SPADE:
> >
> > We thank you for the additional studies, we have added an ablation study for SPADE. We didn’t include it in the first version because the pixel-wise reweighting only considers the location information (e.g. x,y) and cannot incorporate the additional embedding information learned from contrastive learning. As expected, the performance is similar or even worse than standard ResNet as shown below.
> >
> > Here are the results for SPADE compared with standard training and our proposed method and we will update to our table 2,3 in the next version.
> >
> >
> > |**Method**|**buo** |**supply rate**|**buo+supply rate**|**feed**|**kill**|**feed+kill**|
> > |:--|:--|:--|:--|:--|:--|:--|
> > |ResNet|15.8e-2 ± 1.12e-2|4.66e-2 ± 0.13e-2|4.05e-1 ± 1.66e-1|3.49e-3 ± 0.56e-4|1.53e-3 ± 0.44e-3|4.62e-3 ± 0.29e-3|
> > |ResNet+SPADE|14.6e-2 ± 1.62e-2|4.80e-2 ± 0.31e-2|3.67e-1 ± 0.42e-1|5.71e-3 ± 3.08e-4|3.20e-3 ± 1.27e-3|4.61e-3 ± 2.91e-3|
> > |ResNet+SALEM(ours)|**9.06e-2 ± 2.77e-2**|**3.12e-2 ± 0.84e-2**|**1.60e-1 ± 0.73e-2**|**2.28e-3 ± 1.87e-4**|**1.08e-3 ± 0.36e-3**|**3.81e-3 ± 2.22e-3**|
> >
> >
> >
> > $\bullet$  Impact of deeper backbones:
> >
> > While we believe the underfitting-overfitting trade off extends to dynamical system applications, our work serves primarily as a proof of concept in this novel domain. As the first exploration of its kind, our focus is on establishing the viability of the approach rather than exhaustively investigating all possible variations.
> >
> > $\bullet$ regularization technique details:
> >
> > For contrastive learning, we simply use a constant $\lambda$ for covariance loss, as we revealed in the equation 4. For dynamical system learning, there’s no regularizer. In our case, the contrastive learning and dynamical system learning are relatively easy to train, and we will provide code for future users.
> >
> > $\bullet$ Loss combination:
> >
> > Both our contrastive loss (defined by two ratios) and covariance loss (defined by normalized features) are “normalized” in a certain way, therefore, the actual physical value/scale of the problem does not change the loss function. The covariance loss is inspired from Ref [5] which also only has empirical results. We use it because we intuitively thought it could address our dimensional collapse issue and it does. But we don’t have any theoretical justification for combining the losses.
> >
> > Ref 5: VICReg: Variance-Invariance-Covariance Regularization for Self-Supervised Learning, ICLR 2022
> >
> >
> > $\bullet$ Training time:
> >
> > We thank you for the suggestions, we measure the time difference (averaged over 2 runs) for running the training for standard method vs our meta learning framework. For vector-based system (LV system), it takes 86.1 seconds for contrastive learning, 108.4 seconds for dynamics learning with learned embedding, 105.6 seconds for standard dynamics training. On the grid-base examples (vorticity), it takes 2114 seconds for contrastive learning and 12069 seconds for dynamics learning. For standard training methods, it takes 11383 seconds. Overall the contrastive learning part does take a certain amount of time, while the meta dynamics learning is only slightly slower than the standard method. We will update the results in the next version of this paper.

---

### Official Review · Reviewer_9SqM · 2024-11-03

**Soundness:** 2
**Presentation:** 2
**Contribution:** 2
**Rating:** 5
**Confidence:** 3

**Summary:**

This paper proposes to use contrastive unsupervised learning for meta dynamical-system learning.
The authors claim that the method can unsupervisedly identify the system's coefficients and achieve zero-shot forecasting.
Experiments demonstrate the efficacy of the proposed method.

**Strengths:**

- The authors introduce dynamical-system-specific modules for contrastive learning, the local linear least square feature extractor and the spatial adaptive linear modulation. These are simple and may be useful for other dynamical system learning.
- The experiments demonstrate that the proposed method achieves better accuracy than the baselines and seems to be more stable. Showing learning curves may strengthen the latter.

**Weaknesses:**

- It is unclear why the proposed contrastive-learning approach enables unsupervised coefficient identification and zero-shot forecasting.
In particular, I could not understand how and when the model could learn organized embeddings like Figure 3.
It would be interesting if the authors could characterize the dynamical systems that the proposed method could handle.

**Questions:**

- What is $*$ in equation 9?

---

> ### Author Response · Authors · 2024-11-19
> **Rebuttal reply to Reviewer 9SqM**
>
> Dear reviewer:
>
> Thank you for bringing out the question and we are glad to clarify.
>
> $\bullet$ Equation 9:
>
> “*” is just the multiplication sign, we use it to emphasize that this is element-wise multiplication.
>
>
> $\bullet$ Explanation on how contrastive learning works:
>
> In general, with the knowledge of trajectory categorization (e.g. which trajectories below to same system/different systems), we use contrastive learning to learn the encoder which maps a truncation of trajectory (multiple concatenated system states) to the system embedding. This idea is identical with the classic contrastive learning application in the computer vision field. The learned embedding contains the system coefficient information. In the forecasting stage, given a few past state observations (e.g $\{x_{t-3},x_{t-2},x_{t-1}\}$), we first use the encoder to infer the embedding, and the embedding is later used in the dynamics model to infer the system dynamics. Taking the true system function $dx/dt=f_\phi(x)$ as an example, where $x$ is the state and $\phi$ is the system coefficient. We know the output $dx/dt$ should be dependent on both $\phi$ and $x$, therefore a lump model only taking $x$ cannot identify the output accurately. In our method, the encoder is used to infer the embedding $z$ contains the information of $\phi$, and another dynamical model taking both $x$ and $z$ is used to infer $dx/dt$.
>
> $\bullet$ Regarding your question on the organized embedding like figure 3:
>
> To clarify, our unsupervised learning does not ‘’identify’’ the actual coefficient, it aims to learn a representation of the coefficients. This representation could be a linear transformation or more complex transformation. In these two problem setups, it turns out that the rotated linear transformation is the easiest way for the model to describe the embedding, therefore it looks organized and ordered. We also tried to decrease the “number of coefficient sets” and see the results, for these two problems, once we have more than 4 coefficient variants per dimension, the output is always organized. Consider the 1D case with only 3 coefficient sets (e.g. 1,2,3), the model cannot differentiate the magnitude because it only knows 1,2,3 are from different groups, therefore the learned embedding could be ordered as (1,3,2) instead of organized (1,2,3) or (3,2,1).
>
> However, despite the mapping being non-linear (e.g. cubic function) or even non-monotonic (e.g. square functions), as long as the encoder contains useful information regarding the system coefficients and the testing coefficients are not too far away from training distribution, it will always benefit the forecasting stage.
>
> $\bullet$ Regarding the question on what dynamical system our proposed method can “handle”:
>
> We are sorry that we have an ambiguous understanding of “handle” here. If you mean “generate the organized embedding like Figure 3”, when we have two varying coefficients and they vary independently, and the number of coefficients set per dimension is not too small, we can capture that. If you mean “benefit the forecasting”, as we discussed above, if the encoder contains useful information, then it will improve forecasting performance.

---

> > ### Comment · Reviewer_9SqM · 2024-11-26
> >
> > Thank you for the clarification.

---

### Official Review · Reviewer_YjJZ · 2024-11-06

**Soundness:** 2
**Presentation:** 2
**Contribution:** 2
**Rating:** 3
**Confidence:** 4

**Summary:**

This paper points out that it requires a novel method to apply contrastive learning to dynamical systems. Accordingly, the authors developed a contrastive meta-learning method applicable to various dynamical systems. It allows the zero-shot meta-learning on previously unobserved dynamics. This paper makes several contributions, including a novel loss function (e.g., Element-wise Square Ratio Loss (ESRL)) and other learning techniques (e.g., the covariance-based regularization, local linear feature extraction).

**Strengths:**

- This paper introduces a novel problem of contrastive learning on dynamics systems with the perspective of meta learning.

- Limitations on the existing meta learning methods on dynamical systems are clearly pointed out. For example, as the author mentioned in L102, many of them are relying on supervision from system coefficients or few shot adaptation, limiting the practical applications. In this light, the problem statement made by the authors are reasonable.

- Overall the paper is easy to understand and follow. In particular, the effectiveness of contrastive learning is qualitatively well presented in Figure 3, showing that learned embeddings on different trajectories with different coefficients are clearly distinguishable.

**Weaknesses:**

- While the proposed loss function and the covariance-based regularizer demonstrates lower errors than other popular contrastive losses (e.g., Info-NCE, Triplet), technical contributions are not significant. Except it considers element-wise comparison in embedding space, Element-wise Square Ratio Loss (ESRL) has the almost same form with the original Square Ratio Loss (SRL). I don’t think the extension of SRL to the element-wise version is necessarily a novel contribution. The covariance-based regularization is also very well known, as the authors mentioned it is inspired from Bardes et al. (2021). Lastly, I consider local linear square feature extraction as simply linear approximation on the input space.

- I’m not sure the unsupervision of not providing any coefficients to the models is rare and interesting problem setup in dynamical systems. While we can model better trajectory predictors using such knowledge on the systems, recent trajectory prediction models (e.g., neural ODE or RNN) are often solely trained from trajectory and perform pretty well on such synthetic systems. Basically, I disagree with the statement in L441-444 since many previous works still assume the system coefficients are unknown.

- Since the authors explore the meta learning problem, the experimental results should focus on how the proposed learning method improves generalizability on previously unobserved dynamics. For example, the model is trained on trajectories on a set of coefficients and then evaluated on them on a different set of coefficients. However, I couldn’t find such details in Table 1, 2, and 3. Overall, the experiment section is not detailed enough to support L171: "In the context of meta dynamical system learning, our goal is to develop a model that can generalize
from a set of training systems ϕtrain to new, unseen test systems with properties ϕtest."

**Questions:**

N/A

---

> ### Author Response · Authors · 2024-11-19
> **Rebuttal reply to Reviewer YjJZ**
>
> Dear reviewer:
>
> Thank you for the insightful review, while we agree with most of them, we find there could be some misunderstandings and we hope to clarify below.
>
> $\bullet$ element-wise loss function:
>
> The major contribution of our element-wise design of the contrastive loss is to prevent certain dimensions to be “constant”; this cannot be achieved by other loss functions (SRL,triplet,Info-NCE) as they directly compare two vector embeddings. Let us consider a simple case with three examples [[1,0],[2,0],[3,0]], where the second dimension faces dimensional collapse. For SRL, Triplet or Info-NCE, the loss function will be identical to [[1],[2],[3]] with fewer embedding dimensions. However, with our element-wise design, it will evaluate the 2nd dimension individually, and penalize the loss function by moving  the denominator close to 0. In this extreme case the value will be $\frac{0}{0}$, while it will be a large value before reaching this numerical error.
>
> $\bullet$ covariance-based regularization:
>
> Yes, we agree with you that we directly use it from the cited paper for our purpose.
>
> $\bullet$ local linear square feature extraction:
>
> We might also have some misunderstanding here. It is not a linear approximation of the input space (e.g. outputs Wx with parameterized W and input x). Instead, it tries to estimate the system state function (e.g. $min_A ||Ax-\dot{x}||$) and use A to infer the  system coefficient. This treatment is inspired by classical system identification,  where the linear model is the to-go basis function to use and its learned variable directly contains the physical properties (system coefficients). Actually, even for linear problems, the mapping from the trajectory of x to A is quite complex. Therefore, using a directly deep learning method (RNN or MLP) cannot easily capture the features and thus empirically performs worse.
>
> $\bullet$ unsupervision of system coefficients:
>
> Yes we agree with your comment that many previous work assume the system coefficients are unknown. Although there are indeed multiple coefficients/systems, the users treat it as a single lump model and do not explicitly differentiate the systems. Differently, our approach explicitly learns the inter-system difference and uses it in forecasting. There are three folds of benefits: 1. Better interpretability as we explicitly utilize the inter-system information 2. Better forecasting accuracy because of the explicit system coefficients are indeed useful for forecasting (consider system state function f(e,x), if we know environment e, it will improve regression accuracy by a lot) 3. We also empirically showed the performance gain over vanilla neuralODE. As a distant analogy, our method vs lump method is like contrastive pretrain + supervised fine-tune vs pure supervised learning in computer vision. We incorporated the “class”/”category” information in training so we have better performance.
>
> To clarify our claim in line 441-444 “Previous research …, assuming the system coefficients are already known”, by ”previous research” we specifically mean the previous works in meta dynamical system learning that explicitly differentiate inter-system differences. While in the majority of system identification/dynamical learning papers, most of the papers do not differentiate that. This is consistent with your impression. We will update the wording into “Previous research in meta dynamical system learning” in the next version to make it clear.
>
> $\bullet$ missing experiment details:
>
> We delayed the experiment details and coefficient distribution of table 1,2,3 to appendix A: A.1 for dual spring mass system, A.2 for LV system, A.3 for fluid system, A.4 for GS system. Throughout all experiments, the testing environment is unseen to the training process, therefore, we claim “generalize from a set of training systems ϕtrain to new, unseen test systems with properties ϕtest”.

---

> > ### Comment · Reviewer_YjJZ · 2024-11-29
> >
> > Dear Authors,
> >
> > I apologize for the late response. I have carefully read the rebuttal and appreciate the authors' efforts to address my concerns. However, I still couldn't find novel enough technical contributions so decided to stay in the same rating (3).
> >
> > Best regards,
> >
> > Reviewer YjJZ

---

### Meta-Review · Area_Chair_hD3B · 2024-12-19

**Metareview:**

The paper claims to address the problem of applying contrastive learning to dynamical systems in a meta-learning context. It proposes a contrastive meta-learning method applicable to various dynamical systems, enabling zero-shot meta-learning on previously unobserved dynamics. The reviewers raised multiple concerns, including the novelty, types of dynamical systems, some other issues, such as robustness to noise in derivative approximation, application to more complex systems.

After the rebuttal, several key concerns from the reviewers remained unresolved. Some reviewers still didn't find the technical contributions novel enough despite the authors' clarifications. Additionally, the diverged reviews with some reviewers firmly maintaining their concerns after the discussion and rebuttal process suggest that the paper doesn't meet the required standards for acceptance, thus leading to a decision to reject the paper.

**Additional Comments On Reviewer Discussion:**

Diverged reviews and some reviewers pointed out that their concerns were not resolved yet. The positive reviewers does not argue for accept.

---

### Decision · Program_Chairs · 2025-01-22

Reject